# Mortality and Transfusion Requirements in COVID-19 Hospitalized Italian Patients According to Severity of the Disease

**DOI:** 10.3390/jcm10020242

**Published:** 2021-01-11

**Authors:** Elvira Grandone, Raffaele Pesavento, Giovanni Tiscia, Antonio De Laurenzo, Davide Ceccato, Maria Teresa Sartori, Lucia Mirabella, Gilda Cinnella, Mario Mastroianno, Lidia Dalfino, Donatella Colaizzo, Roberto Vettor, Angelo Ostuni, Maurizio Margaglione

**Affiliations:** 1Thrombosis and Haemostasis Unit, Fondazione I.R.C.C.S. “Casa Sollievo della Sofferenza”, 71013 San Giovanni Rotondo, Italy; g.tiscia@operapadrepio.it (G.T.); antonio.delaurenzo@operapadrepio.it (A.D.L.); d.colaizzo@operapadrepio.it (D.C.); 2Ob/Gyn Department of the First I.M. Sechenov Moscow State Medical University, 119435 Moscow, Russia; 3Department of Internal Medicine, University of Padua, 35122 Padua, Italy; r.pesavento@unipd.it (R.P.); ceccatodavide@gmail.com (D.C.); mtsartori@unipd.it (M.T.S.); roberto.vettor@unipd.it (R.V.); 4Department of Anesthesia and Intensive care, University of Foggia, 71122 Foggia, Italy; l.mirabella@unifg.it (L.M.); g.cinnella@unifg.it (G.C.); 5Scientific Direction, Fondazione I.R.C.C.S. “Casa Sollievo della Sofferenza”, 71013 San Giovanni Rotondo, Italy; m.mastroianno@operapadrepio.it; 6Anesthesia and Intensive care Unit, University of Bari, 70121 Bari, Italy; lidia.dalfino@yahoo.com; 7Immunohematology and Transfusion Medicine Service, Azienda Ospedaliero-Universitaria Consorziale Policlinico di Bari, University of Bari “Aldo Moro”, Struttura Regionale Coordinamento, 70124 Bari, Italy; angelo.ostuni@policlinico.ba.it; 8Medical Genetics, University of Foggia, 71122 Foggia, Italy; m.margaglione@unifg.it

**Keywords:** COVID-19, mortality, transfusion, medical ward, ICU

## Abstract

There is paucity of data on the transfusion need and its impact on the overall mortality in patients with COVID-19. We explored mortality in hospitalized patients with COVID-19 who required transfusions. Information on clinical variables and in-hospital mortality were obtained from medical records of 422 patients admitted to medical wards or the Intensive Care Unit (ICU). In-hospital mortality occurred in 147 (34.8%) patients, 94 (63.9%) of whom were admitted to the ICU. The median fatalities age was 77 years (IQR 14). Overall, 100 patients (60 males) received transfusion during hospitalization. The overall mortality was significantly and independently associated with age, ICU admission, Chronic Kidney Disease (CKD), and the number of transfused Red Blood Cell (RBC) units. Specifically, CKD was associated with mortality in patients admitted to medical wards, whereas the number of transfused RBC units predicted mortality in those admitted to the ICU. Transfusion strongly interacted with the admission to ICU (OR: 9.9; 95% CI: 2.5–40.0). In patients with COVID-19, age is one of the strongest risk factors in predicting mortality independently of the disease’s severity. CKD confers a higher risk of mortality in patients admitted to medical wards. In those admitted to the ICU, the more RBC units are transfused, the more mortality increases.

## 1. Introduction

The SARS-CoV-2 disease outbreak has caused a dramatic rise in mortality worldwide. Concomitantly, the pandemic has had a significant impact on blood supplies through a reduction in blood donation [1,2]. In this context, it appears of relevance to understand the real need of transfusion in patients with COVID-19—the respiratory disease consequent to the infection—and its potential impact on clinical outcomes, first of all mortality.

Indeed, COVID-19 can present with a variety of clinical manifestations (from nil, to pauci-symptomatic, to dramatic disease), leading to mortality especially those subjects with one or more comorbidities [3]. The mortality rate is particularly high in patients admitted to the intensive care unit (ICU) [4,5], with older patients showing a higher fatality rate [5].

Patients admitted to the ICU have generally a higher need of blood transfusion than those admitted to medical wards [6]. Anemia in COVID-19 is uncommon on admission and especially during the first 3 days of hospitalization [1]. The available data show a high rate (78.5%) of transfusion in those admitted to the ICU [7], whereas the general prevalence in hospitalized patients is similar to that reported in other settings of patients (1.2%) [8,9].

Patients with COVID-19 are often elderly and show comorbidities. It has been hypothesized that in the presence of respiratory symptoms and need of ventilation, these patients can benefit from Red Blood Cell (RBC) transfusions to maintain hemoglobin levels above 70 g/L [1]. However, specific transfusion trial data on patients with COVID-19 are still lacking and physicians follow general recommendations to transfuse these patients [1].

The purpose of this retrospective multi-center study was to explore mortality and comorbidities in hospitalized patients with COVID-19 who required transfusions. We investigated the role of potential risk factors for mortality in critically ill vs. non-critically ill patients.

## 2. Patients and Methods

We retrospectively recruited 422 patients, 179 of whom were admitted to the ICU, with a laboratory-confirmed diagnosis (i.e., RT-PCR according to the protocol established by the WHO) and radiologically confirmed pneumonia observed in four Italian academic hospitals (University hospital of Padova, Research Institute “Casa Sollievo della Sofferenza”, University hospital of Foggia, and University hospital of Bari) from 3 March until 30 August 2020.

## 3. Data Collection

Demographic data, comorbidities, medications, and clinical variables including respiratory support and in-hospital mortality were obtained from medical records.

All patients received treatment with multiple drugs (i.e., hydroxychloroquine, heparins, corticosteroids, antivirals, anti-interleukin-6 receptor, antibiotics, vasopressors, and invasive or non-invasive ventilation). We identified all types of transfusion products used during the hospitalization. Furthermore, we divided transfusion events into three subgroups according to the transfusion products: RBC, platelet concentrate, and plasma.

The proportions of associated risk factors and clinical outcome parameters in critically ill and non-critically ill patients were compared.

The study was approved by the local Review Board and carried out in accordance with the Declaration of Helsinki.

## 4. Statistical Analysis

Discrete variables are summarized as median (IQR) values. Categorical variables are presented as whole numbers with percentages. We used the χ^2^ test, Fisher’s exact test, or Mann–Whitney test to compare differences where appropriate. Multivariable logistic analysis was performed to assess the independent association of all-cause mortality with significant variables found at univariate analysis. Adjustment was made for the following confounders: Age, sex, comorbidities, ICU admission, medical therapy, transfusion, number of transfused RBC units, and hemoglobin values at admission. All statistical procedures were performed using the SPSS 25.0 software (SPSS Inc., Chicago, IL, USA).

## 5. Results

### 5.1. Patient Characteristics

A slightly higher percentage (59%) of males was observed among the whole cohort of enrolled patients (Table 1). Overall, 179 (42.4%) out of 422 patients were admitted to the ICU, most of them with at least one comorbidity. The vast majority received a Low-Molecular Weight Heparin (LMWH) at prophylactic doses (i.e., enoxaparin 4000 U/day subcutaneously), as shown in Table 1.

The standard prophylactic LMWH dose was labeled as the administration of enoxaparin 4000 IU once daily; intermediate doses as 60 mg subcutaneously once daily, or 4000 IU twice daily; therapeutic doses as the administration of 100 U/Kg twice daily. 

### 5.2. Transfusion Rates and Factors Associated with Transfusion 

Overall, 100 patients (60 males; median age 73 years; IQR 15.8) needed transfusion during hospitalization (Table 1). All but two received RBC units, whereas a minority received plasma (*n* = 11) or platelets (*n* = 9) in addition to RBC units. Furthermore, 62% received one to three RBC units and 20% from four to six units. Features of non-transfused patients vs. those who received the RBC units are shown in Table 2.

Patients administered with LMWH (with/wo antiplatelet therapy) were not more often transfused than those who did not receive antithrombotic drugs. Those who were administered with intermediate or therapeutic doses of LMWH did not receive a higher number of RBC units (median 0; IQR 1) than those administered with prophylactic doses (median 0; IQR 1) (Mann-Whitney test; *p*: N.s.).

Hemoglobin levels at admittance to the hospital was negatively correlated with the number of RBC units transfused (Pearson; *p* < 0.001).

### 5.3. Factors Affecting Mortality

In-hospital mortality occurred in 147 (34.8%) patients, 53 (21.8%) of whom were admitted to medical wards (Figure 1). The median fatalities age was 77 years (IQR 14).

At the univariate analysis, mortality was significantly higher in those who were transfused (*n* = 62, 62%) vs. those who were not transfused (*n* = 85, 26.4%) (Fisher’s exact test; *p* = 0.000). All but one patient who also needed plasma and/or platelets died during hospitalization. Finally, those with lower Hb levels at admittance had a significantly higher risk of in-hospital mortality (g/dl 12.1; IQR 2.8 vs. 13.3; IQR 2.9; Mann-Whitney test; *p* = 0.000).

Patients who underwent non-invasive and invasive ventilation had a higher risk to receive transfusions (OR: 4.7; 95% CI: 2.9–7.6 and OR: 4.1; 95% CI: 2.5–6.8, respectively). The use of hydroxychloroquine and LMWH (with or w/o antiplatelet drugs) did not affect the relationship between transfusion and mortality.

At the logistic regression, correcting for potential confounders, mortality was significantly and independently associated with age, ICU admission, CKD, and the number of transfused RBC units (Table 3).

When we separately analyzed the two groups, we found that CKD was significantly associated with mortality in patients admitted to medical wards, whereas the number of transfused RBC units predicted mortality in those admitted to the ICU.

In addition, the likelihood of receiving transfusions strongly interacted with admission to the ICU (*p*: 0.001; OR: 9.9; 95% CI: 2.5–40.0).

## 6. Discussion

We explored patients’ need for transfusion in patients with COVID-19 hospitalized in medical wards or the ICU and investigated the potential role of several risk factors in predicting mortality in patients admitted to medical wards or the ICU.

We found that age is one of the strongest risk factors in predicting mortality independently of the disease’s severity: The risk rises up to 10% for an increase of each year of age. The immune response decreases and becomes less efficient with the increasing age, therefore the adverse outcomes in older patients are not unexpected [10].

The percentage of patients with CKD in our study does not differ much from that observed in other cohorts of patients with COVID-19 [11]. This condition, in the present study, confers a higher risk of mortality, as already shown [12,13]. Indeed, from 11.6% (present study) to 16.2% [11] of patients with COVID- 19 show CKD and the lower the estimated glomerular filtrate rate (eGFR), the higher the risk of mortality (eGFR 30–59 mL/min/1.73 m^2^: HR 3.01; eGFR < 30 mL/min/1.73 m^2^: HR 6.61) [12].

The ICU access is a strong and independent predictor of mortality (OR 5.09). The present data show that critical illness is significantly associated with a higher need of transfusion. Indeed, the percentage of transfused patients was significantly higher in those admitted to the ICU (41.9%) than in those admitted to medical wards (10.3%). This is not surprising, as transfusion requirements progressively increase with the worsening of the clinical condition. Consistent with these findings, patients undergoing non -invasive and invasive ventilation are significantly more transfused than those who do not need (non-invasive or invasive) ventilation (Table 2). However, when we corrected these associations for potential confounders (i.e., age, sex, comorbidities, ICU admission, medical therapy, transfusion, number RBC units, hemoglobin at admission), we found that non-invasive, as well as invasive ventilation were not independently associated with mortality; which is likely due to the fact that most patients admitted to the ICU undergo ventilation (Table 1).

Our data confirm and extend those observed in other settings [14]. Indeed, hemoglobin levels at admittance to the hospital were negatively correlated with the number of RBC units transfused (Pearson; *p* < 0.001), although the logistic regression did not show an independent and significant association between this parameter and mortality.

We found that in patients admitted to the ICU, the number of RBC units strongly predicts the overall mortality, which increases by 37% per unit of transfused RBC units. Our data are in line with those by Rim et al., who showed that most of the transfused Korean patients with COVID-19 were hospitalized in the ICU [7]. The strong relationship between transfusion and all-cause mortality described in the Korean population was confirmed in the present study, which makes the data more robust: The more RBC units are transfused, the more mortality increases. Furthermore, our data give support to findings from clinical studies carried out in critically-ill patients [15]. Indeed, a randomized controlled trial and an observational study demonstrated that restrictive transfusion strategies (hemoglobin levels of 7–9 g/dL to trigger transfusion) significantly decreases inpatient mortality [16,17].

## 7. Strength and Limitations

This is a multi-center study, involving four Italian hospitals. We included medical wards, as well as the ICU, thus offering a snapshot of patients with different disease severity. The outcome of the study is robust, as all-cause mortality was included. However, due to the retrospective nature of the study, we acknowledge the likelihood of unknown potential confounders.

## 8. Conclusions

The present data suggest that age, CKD, and ICU access are strong predictors of mortality in Italian patients with COVID-19. Furthermore, the number of transfused RBC units is significantly and independently associated with mortality in patients admitted to the ICU. The design of our study does not allow us to draw conclusions on the cause-effect relationship between transfusion requirements and the severity of the disease. To provide definite information about the cause-effect relationship, prospective studies are needed.

## Figures and Tables

**Figure 1 jcm-10-00242-f001:**
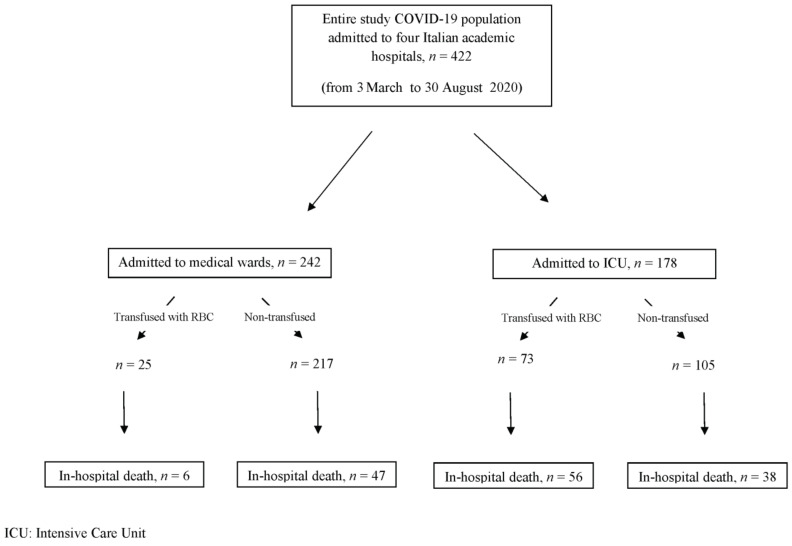
Diagram describing the entire cohort of patients by transfusions requirement and mortality.

**Table 1 jcm-10-00242-t001:** Demographic and clinical information of the entire study population and divided by admission to hospital wards.

Variables	All Patients *n* = 422	Patients in Medical Wards *n* = 243	Patients in ICU *n* = 179	*p*
Sex, male/female	249/173	123/120	126/53	0.000
Age median (IQR)	72 (20)	73 (22)	70 (18)	0.023
Smoking, *n* (%)	24 (5.7)	16 (6.6)	8 (4.5)	n.s.
Diabetes, *n* (%)	88 (20.9)	48 (19.8)	40 (22.3)	n.s.
Hypertension, *n* (%)	207 (49.1)	122 (50.2)	85 (47.5)	n.s.
History of Cancer and/or Active Cancer, *n* (%)	55 (13.1)	34/3 (15.2)	18/0 (10.1)	n.s.
Cerebrovascular disease, *n* (%)	25 (5.9)	10 (4.1)	15 (8.4)	n.s.
Cardiovascular disease, *n* (%)	107 (25.4)	65 (26.7)	42 (23.5)	n.s.
Chronic kidney disease, *n* (%)	49 (11.6)	30 (12.3)	19 (10.6)	n.s.
Chronic Obstructive Pulmonary Disease, *n* (%)	46 (10.9)	24 (9.9)	22 (12.3)	n.s.
**Antithrombotic Therapy**				
Anticoagulants at admission, *n* (%)	53 (12.6)	37 (15.2)	16 (8.9)	n.s.
Antiplatelets at admission, *n* (%)	91 (21.6)	51 (21.0)	40 (22.3)	n.s.
**LMWH during hospitalization** No prophylaxis, *n* (%) Prophylactic doses, *n* (%) Intermediate doses, *n* (%) Therapeutic doses, *n* (%)	78 (18.5) 282 (66.8) 39 (9.2) 23 (5.5)	50 (20.6) 180 (74.1) 9 (3.7) 4 (1.6)	28 (15.6) 102 (57.0) 30 (16.8) 19 (10.6)	0.000
LMWH+ antiplatelets drug during hospitalization, *n* (%)	74 (17.5)	40 (16.5)	34 (19)	n.s.
Major or NMCR Hemorrhage	15 (3.5)	7 (2.9)	8 (4.5)	n.s.
Transfusion (RBC/Plt/Plasma)	100	25 (10.3)	75 (41.9)	0.000
RBC median (IQR)	0 (0)	0 (0)	0 (2)	0.000
Hb at admission median (IQR)	12.7 (3.1)	12.9 (2.95)	12.4 (3.3)	0.039
Death during hospitalization (%)	147 (34.8)	53 (21.8)	94 (52.5)	0.000
**COVID-19 treatment ***				
Hydroxychloroquine	145 (34.4)	49 (20.2)	96 (53.6)	0.000
Ritonavir/lopinavir	62 (14.7)	21 (8.6)	41 (22.9)	0.000
Antibiotics	247 (58.5)	128 (52.7)	119 (66.5)	0.005
Steroids	83 (19.7)	43 (1.2)	40 (22.3)	0.000
Non-Invasive Ventilation	118 (28)	29 (11.9)	89 (49.7)	0.000
Invasive Ventilation	44 (10.4)	0	44 (24.6)	0.000

Categorical variables are expressed as numbers and percentages; continuous variables are expressed as the mean (± standard deviation) or median (IQR). ICU: Intensive Care Unit; LMWH: Low-Molecular Weight Heparin; NMCR: Non-major clinically relevant; RBC: Red blood cells; Plt: Platelets; * Some data missing.

**Table 2 jcm-10-00242-t002:** Clinical features of patients with COVID-19: Differences between patients transfused with red blood cell (RBC) units and non-transfused patients.

	Non-Transfused *n* = 322	RBC Transfusions *n* = 98	*p*
Sex, male/female	189/133	58/40	n.s.
Age median (IQR)	72 (21)	73 (15.5)	n.s.
Smoking, *n* (%)	21 (6.5)	3 (3.1)	0.04
Hb at admission median (IQR)	13.3 (2.5)	10.7 (2.8)	<0.01
Diabetes, *n* (%)	65 (20.2)	23 (23.5)	n.s.
Hypertension, *n* (%)	159 (49.4)	48 (49)	n.s.
History if Cancer and/or Active Cancer, n (%)	46 (14.3)	9 (9.2)	n.s.
Cerebrovascular disease, *n* (%)	19 (5.9)	6 (6.1)	n.s.
Cardiovascular disease, *n* (%)	75 (23.3)	32 (32.7)	n.s.
Chronic kidney disease, *n* (%)	35 (10.9)	14 (14.3)	n.s.
Mechanical Ventilation	46 (14.3)	54 (55.1)	<0.01
Invasive Ventilation	57 (17.7)	43 (43.9)	<0.01
Chronic Obstructive Pulmonary Disease, *n* (%)	34 (10.6)	12 (12.2)	n.s.
**LMWH during hospitalization**			
No prophylaxis *n (%)* Prophylactic doses, *n (%)* Intermediate doses, *n (%)* Therapeutic doses, *n* (%)	56 (17.4) 223 (69.3) 29 (9.0) 14 (4.3)	22 (22.4) 59 (60.2) 10 (10.2) 9 (9.2)	n.s.
LMWH+ antiplatelets drug during hospitalization, *n* (%)	56 (17.4)	17 (17.3)	n.s.
Major or NMCR Hemorrhage, *n* (%)	12 (3.7)	3 (3.1)	n.s.

RBC: Red blood cells.

**Table 3 jcm-10-00242-t003:** Logistic regression—factors affecting mortality in all patients and according to admission to the ICU or medical ward.

All Patients			
**Variable**	***p***	**OR**	**95% CI**
Age	0.00	1.08	1.05–1.10
ICU access	0.00	5.09	2.86–9.05
CKD	0.01	2.51	1.22–5.18
No. of transfused RBC units	0.00	1.35	1.15–1.60
Patients admitted to the ICU			
**Variable**	***p***	**OR**	**95% CI**
Age	0.001	1.06	1.02–1.09
No. of transfused RBC units	0.017	1.37	1.06–1.77
Patients admitted to medical wards			
**Variable**	***p***	**OR**	**95% CI**
Age	0.000	1.09	1.05–1.13
CKD	0.005	4.19	1.54–11.37

CKD: Chronic kidney disease; RBC: Red blood cells; ICU: Intensive Care Unit.

## Data Availability

The data presented in this study are available on request from the corresponding author.

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
