# Peer review of "Mortality and Transfusion Requirements in COVID-19 Hospitalized Italian Patients According to Severity of the Disease"

_jcm, 2021, doi:10.3390/jcm10020242_

Round 1
Reviewer 1 Report
- "tranfsuion requirement" should be described in the methods : 3 patients did not received RBC, is it relevant to include them in tranfsusion requirement?
- is cancer history correlated to tranfusion requirement? Underlying condition may explain severity of disease and anemia, and it is a possible biais.
- Based on this study, we can conclude that tranfusion requirement is associated with higher mortality. However there is no data that led us to conclude that a restrictive threshold should be used. Conclusions must be cautious.
Author Response
We would like to thank this reviewer for kind appreciation of our work and the chance he/she is giving us to improve its quality.
Detailed answers to comments:
"transfusion requirement" should be described in the methods : 3 patients did not received RBC, is it relevant to include them in transfusion requirement?
Response:
In the methods’ section we now describe “transfusion requirement” ( page 4, line 17-19). Indeed, we state as follows: “We identified all types of transfusion products used during the hospitalizations. Furthermore, we divided transfusion events into three subgroups according the transfusion products: Red Blood Cells (RBC), platelet concentrate, plasma.” Furthermore, in table 1 it is now detailed that we are reporting all transfusion products, whereas table 2 in its present form illustrates features of those patients who were transfused only with RBC units.
In the results’ paragraph ( page 5, line 19-20) we report that table 2 illustrates data of those who received RBC.
is cancer history correlated to transfusion requirement? Underlying condition may explain severity of disease and anemia, and it is a possible bias.
Response:
We thank this reviewer for the proper observation. Indeed, the percentage of transfused patients with cancer was higher than that of non- transfused ones ( 14.3% vs 9.2%). However, this percentage did not reach significance, likely due to the sample size.
As pointed out in the discussion’s section ( page 8, line 1-4), haemoglobin levels at admittance to hospital were negatively correlated with the number of RBC units transfused (Pearson p<0.001), although the logistic regression did not show an independent and significant association between this parameter and mortality.
Based on this study, we can conclude that transfusion requirement is associated with higher mortality. However, there is no data that led us to conclude that a restrictive threshold should be used. Conclusions must be cautious
Response:
We recognize that we must be more cautious in conclusions, as present data do not support the concept that a restrictive threshold should have an impact on the outcome. Consistent with this, we erased the last sentence.
Reviewer 2 Report
- Discussion. The relation among critical illness, transfusion and fatality should be commented. Do transfusion requirements just indicate the severity of illness or do transfusions contribute to the severity making things worse for the patient? The fact that "...critically illness is significantly associated with higher need of transfusion" is mentioned, however the reasons for bad outcomes are not clarified and therefore the whole point of the paper is a little obscure.
- Diagrammes or figures showing critical data of the paper would greatly enrich the results' presentation.
- Some expressions could be better stated, e.g. "fatalities" instead of "dead patients" (L. 26, 113), "transfusions" instead of "transfusion" (L. 27) since there were more than one transfused patients.
Author Response
General comments
We would like to thank this reviewer for the effort to revise our manuscript. As suggested, we have implemented the introduction. We have added some sentences ( page 3, lines 15-16 and 19-23), highlighting the available data on anaemia and transfusion needs in COVID-19 patients.
Detailed answers to comments:
Discussion. The relation among critical illness, transfusion and fatality should be commented. Do transfusion requirements just indicate the severity of illness or do transfusions contribute to the severity making things worse for the patient? The fact that "...critically illness is significantly associated with higher need of transfusion" is mentioned, however the reasons for bad outcomes are not clarified and therefore the whole point of the paper is a little obscure.
Response:
We thank the reviewer for this interesting and challenging comment. We would like to point out that logistic regression suggests that RBC transfusion is associated with admission to ICU independently of other factors ( i.e. comorbidities, ventilation, etc). The design of our study limits our ability to determine whether transfusion requirements just indicate the severity or contribute to the severity of the disease. Indeed, retrospective studies may not provide definite information about cause-effect relationship (Hess DR. Respir Care 2004;49(10):1171–1174. ). We need prospective studies to clarify reasons of the bad outcomes.
This relevant comment was added at the end of the Conclusions ( page 8, line 24-27).
Diagrammes or figures showing critical data of the paper would greatly enrich the results' presentation.
Response:
We thank for this suggestion and accordingly are showing a flow diagram showing critical data (Figure 1)
Some expressions could be better stated, e.g. "fatalities" instead of "dead patients" (L. 26, 113), "transfusions" instead of "transfusion" (L. 27) since there were more than one transfused patients.
Response:
As suggested, we changed “dead patients” in “fatalities” and “transfusion” in “transfusions”( Abstract page 2, line 13; Results page 6, line 5).

Round 2
Reviewer 1 Report
this original study provides useful information